# Text2whaiora after a suicide attempt: Text message design alongside people with lived experience

Lillian Ng [1,2]* , Danielle Diamond[2] , Mike Ang[2]

1 Department of Psychological Medicine, Faculty of Medical and Health Sciences, The University of Auckland, Auckland, New Zealand, 2 Department of Mental Health and Addictions, Health New Zealand Te Whatu Ora Counties Manukau, Auckland, New Zealand

☯ These authors contributed equally to this work.

* lillian.ng@auckland.ac.nz

## Abstract

### Background

People with lived experience have had limited opportunities to meaningfully contribute to the design of caring contacts interventions. The objective of this study was to co-design text messages with peer support specialist staff to determine optimal language and delivery, within a cultural context.

### Methods

In this qualitative study, participants were professional peer support specialist staff with lived experience employed by specialist mental health services. They were asked to evaluate the initial series of text messages by taking part in a focus group using a semi-structured interview. This was audiotaped, transcribed and analysed using reflexive thematic analysis with specific coding of cultural themes.

### Results

Three main themes were identified: upholding a person's autonomy; establishing connection as a bridge to safety; and, words as healing rongoā (remedy). The last theme contained a cultural subtheme: Māori language providing entry to the Māori world.

### Conclusion

People with lived experience breathe empowerment and hope into caring contacts interventions and should be considered vital partners in developing any suicide prevention initiative. Feeling genuinely cared for promotes connection and may enable an internal sense of safety. Tailoring of texts can be enhanced by culturally nuanced language.

**Data Availability Statement:** Data cannot be shared publicly because the participants have been users of mental health services. Excerpts of the transcripts relevant to the study will be made

available upon reasonable request from the corresponding author. Data requests may be sent to the New Zealand Health and Disability Ethics Committee by e-mail: hdecs@health.govt.nz or Professor Trecia Wouldes, Head of Department of Psychological Medicine, Faculty of Medical and Health Sciences: t.wouldes@auckland.ac.nz

**Funding:** Lillian received an Oakley Mental Health Foundation Grant 3725210 oakleymentalhealth.co.nz The funders did not play any role in the study design, data collection and analysis, decision to publish, or preparation of the manuscript.

**Competing interests:** The authors have declared that no competing interests exist.

## Introduction

SMS text messages are becoming a powerful tool to connect service users to mental healthcare services [1]. They can be sent to a person shortly after they attempt suicide and reach them over a sustained period of time [2]. Text messages are an example of caring contacts [3], an intervention for suicide prevention that is gathering research interest. The essential element of caring contacts is the use of brief, caring language to communicate the sender's awareness of the patient and convey positive feelings towards them [4]. Caring contacts involve sending a person a series of personalised text-based communications that express interest and concern for their wellbeing. The person is not required to respond or act, and the contact may indirectly facilitate engagement with healthcare services [3]. In principle, caring contacts are an asynchronous, non-intrusive, inexpensive intervention with a potentially broad reach [5, 6]. In New Zealand, and globally, SMS text messages are a practical means to enhance help-seeking behaviour and engagement with treatment [7]. Caring contacts can be delivered to people when they are discharged from acute psychiatric settings, such as an Emergency Department (ED) [8].

The concept of caring contacts originated in the form of typewritten letters [3] but progressively evolved to include postcards [9], emails [10] and text messages [11]. The protective effect on preventing suicide may be the recipient of messages developing a sense of connection and relatedness. They may perceive this connection and social support as a buffer against stress and suicidal behaviour. Text messaging has advantages over other types of interventions in being able to reach people wherever they are located and addressing suicide risk during treatment gaps [10]. However, the evidence for efficacy of brief contact interventions is mixed [12]. Findings from a small Australian study show text messages provided comfort to service users in knowing that there was someone who cared [1]. Meta-analytic studies show caring contacts reduce some suicidal behaviours and there is limited evidence in reducing suicide mortality, hospitalisation or presentations to the emergency department [5]. A randomised controlled trial of caring text messages sent to military personnel in the United States had inconsistent results but recipients had less suicidal ideation and fewer attempts at suicide [11].

Caring contacts are culturally and ecologically relevant to indigenous peoples and collaboration with specific communities allows messages to be tailored to their needs [13]. In Aotearoa New Zealand, consideration of Māori health dimensions is essential in working with *tangata whaiora*, defined as 'a person seeking wellness' [14]. Māori models of health provide access to Māori world views, incorporating hinengaro (mental), cultural, wairuatanga (spiritual), environmental and whānau (family) dimensions [15, 16]. Increasing suicide rates among indigenous peoples have been linked to cultural alienation, not helped by western philosophies and traditions [17]. A multi-dimensional approach to suicide prevention is necessary, particularly research that affirms indigenous self-governance, culture and world views. Culturally nuanced findings may be transferable internationally, where there are indigenous or first nations peoples who are familiar with the Turamarama declaration [17], which affirms the agency and rights of these communities to find their own solutions for health, including suicide prevention.

The Lancet Commission recommends that people with lived experience of suicidal behaviour be involved in all stages of treatment development [18]. This is in keeping with the movement toward patient-oriented research, that aims to empower patients in the process of research, optimise research design, enhance validity and make knowledge exchange more effective [19]. The goal of patient-oriented research is achieved when research findings improve the health of the population under study [20, 21]. Having lived experience of a

condition, of navigating the healthcare system and being willing to share those experiences are important contributions.

So far, people with lived experience have had limited opportunities to meaningfully contribute to the design of caring contacts interventions. These studies typically involve small numbers [22, 23]. Larsen et al's research co-designed a SMS brief contact intervention with lived experienced groups for delivery in an ED setting after a suicide attempt. People with suicidal ideation or suicidal behaviour had a preference for SMS, compared with other forms of follow up [22]. In this research, we aimed to co-design SMS text messages with peer support specialist staff working in mental health service to determine optimal language and delivery, tailored to our cultural context. We describe our peer specialist participants as "people with lived experiences," and acknowledge that this term encapsulates a range of experiences such as distress and a range of social stressors [24]. We take "lived experience" to mean someone who has an experience of mental distress, and acknowledge their diverse experiences, and intersections with other identities. In this article, we use Māori terms that can be translated differently in other contexts.

## Methods

### Research context

The research team are located at Health New Zealand Te Whatu Ora Counties Manukau in Auckland, an ethnically diverse city (approximately 1.5 million) which includes New Zealand's largest Māori and Pacific population. At our locality, there are approximately 2000 mental health-related presentations to the emergency department per year. Most who attempt suicide will be discharged to primary care or to a local community mental health team. There can be a significant delay in follow up after the person leaves the emergency department and some will receive minimal follow up, if at all. The researchers' backgrounds include expertise in academic, clinical, quality improvement, policy and suicide prevention coordination. One of the authors (DD), is a psychologist of Māori (Kāi Tahu and Ngāpuhi) descent, and was instrumental in leading the research team in Māori and Pacific consultation to ensure the study incorporated culturally safe practice, equity and the needs of Māori participants in consideration of constitutional Te Tiriti o Waitangi (Treaty of Waitangi) principles. As such, an indigenous lens on the research process, which include specific analysis of cultural nodes and their interpretation in reporting the findings.

Additional information regarding the ethical, cultural, and scientific considerations specific to inclusivity in global research is included in the S1 File.

### Study design and recruitment

This qualitative study was approved by the New Zealand Health and Disability Ethics Committee. We used interpretive description, a methodology that combines a clinical and research lens to study design, which has pragmatic utility in applying findings to clinical contexts [25], including psychiatric settings [26]. The first draft of a series of seven text messages were designed by a psychiatrist (MA) in consultation with a clinical psychologist (Table 1). Consultation was then obtained from the specialty Māori and Pacific cultural teams and the peer support professional lead based at Te Whatu Ora Counties Manukau. The participants were professional peer support specialist staff employed by specialist mental health services, with lived experience of mental health care. Most had direct experience of suicidal behaviour, such as suicidal ideation, a suicide attempt or the death of a family member from suicide. All had experience of supporting someone with suicidal behaviour. They were asked to evaluate the initial series of text messages by taking part in a focus group using a semi-structured interview.

**Table 1. Initial and revised text message series.**

| TEXT SERIES | Original Text | Modified Text |
| --- | --- | --- |
| Text 1 (Within Week 1) | Kia ora [name], It was good to meet you yesterday. If you need more support you can freephone 1737. Arohanui [name] | Tēnā Koe [name], It's [name] here from [location]. I was just wondering how you were going after yesterday. Remember for support freephone or text 1737. Arohanui (warm regards) [name] |
| Text 2 (Week 1) | Kia ora [name], Hope your day is going well. You can freephone 1737 if you want to kōrero about how things are going. Take care [name] | Tēnā Koe [name], [name] here from [location]. Just seeing how you are doing? You are most welcome to freephone or text 1737 if you want to korero(talk) about how things are going. Take care. [name] |
| Text 3 (Week 2) | Tēnā Koe [name], Hope you are feeling uplifted in your Mauri today. 1737 is available on text for support. Ngā mihi [name] | Kia ora [name], We were thinking of you. Hope you are feeling uplifted in your Mauri (energy) today. If you like to you can contact 1737 by free phone or text. Ngā mihi (Regards) [name] |
| Text 4 (Week 4) | Kia ora [name], Hope all is well with you and your whaanau. You can Freephone 1737 at any time if you want a chat. Kia kaha. [name] | Kia ora [name], Hope all is well with you and your whanau (family). You can Freephone or text 1737 at any time if you want a kōrero (chat). Ngā mihi (regards) [name] |
| Text 5 (Week 7) | Kia ora [name], [name] here. Thinking of you and wishing you well. For support or a kōrero Freephone 1737. | Kia ora [name], [name] here. Thinking of you and wishing you well. For support or a kōrero (talk) Freephone or text 1737. |
| Text 6 (Week 11) | Tēnā Koe [name], Hope you are feeling settled in your wairua. Freephone 1737 for support. Ngā mihi [name] | Tēnā Koe [name], Hope you are feeling settled in your wairua (spirits). Freephone or text 1737 for support if you want. Ngā mihi [name] |
| Text 7 (Week 12) | Kia ora [name], Hope all is going well for you. You can Freephone or text 1737 for support or a korero. Arohanui. [name] | Kia ora [name], Hope everything is going well for you. Support or a korero (chat) continues to be here for you. Freephone or text 1737. Arohanui (warm regards) [name] |
| Optional on birthday | Happy Birthday [name], We hope it's been ka pai and that this year brings you good things! From [name] | No change |

The coordinators of the peer support programme invited potential participants to the study who were sent a participant information sheet (PIS), including a copy of the proposed questions, prior to the focus group interview. Written consent was obtained from all participants. The recruitment period of the study was from 28 February 2023 to 28 March 2023.

## Data collection

The researchers met with the participants for two sessions. In the first session (60 minutes), the study rationale and context were explained to participants at which time they had the opportunity to ask questions. In the second session (90 minutes), participants were asked to provide basic demographic information and the focus group interview was audio-recorded.

## Analysis

The audiotaped recording from the focus group interview was professionally transcribed. The transcripts were de-identified and stored in NVivo version 14, a computer assisted qualitative database system. Participants were given the option to check their transcript for accuracy. Data was analysed using reflexive thematic analysis [27, 28]: first steps were familiarisation and engagement with transcripts, descriptive coding (defined as facets of meaning) and generating initial themes (defined as central organising concepts). To enhance analytic rigour, the research team discussed queries regarding context and initial themes with a researcher with expertise in qualitative methodology, who independently coded the transcript. Specific coding of Māori cultural themes was conducted to the analysis (DD) to ensure tino rangatiratanga (ownership) over matauranga Māori (Māori knowledge). In a recursive process, themes were refined in relation to the research question, returning to the original data for deeper interpretation. The process was documented in memoranda in keeping with an audit trail.

## Results

There were seventeen participants, ranging in age from 26 to 57 years, including Māori, Samoan, Niuean, Cook Island, European, Indian and Chinese ethnicities with between 1 to 14 years of experience employed by publicly funded mental health services in peer specialist support roles. Three main themes were identified: 1) upholding a person's autonomy; 2) establishing connection as a bridge to safety and 3) words as healing rongoā (remedy). A cultural subtheme was identified within the third theme: Te Reo Māori (Māori language) providing entry to te Ao Māori (the Māori world).

## 1) Upholding a person's autonomy

This theme was focused on tangata whaiora (person seeking wellness) and their progression from being in an emergency department having just attempted suicide to recovering from the ordeal. Participants highlighted the concept of returning power to the person. They were cognisant of wider effects of suicide on other family members, with associated trauma, suffering and devastation:

*"The family are the ones on the ground floor, ground zero, they're the ones that have the initial impact. I found my older brother and the impact of seeing [him] do that is really destructive. Having some support for the whanau while they're going through that difficult time"*

The participants wanted the text messages to "*create inclusion,*" to include key supports but ensure permission was granted to let family know they were receiving text messages:

*"Can we ask whānau if they would like them to receive them as well?"*

They were mindful of evoking memories of the suicide attempt and carefully holding the person in mind with "*a statement you're making in these texts.*" They endorsed practical encouragement in SMS to "*help them through challenges they're experiencing.*"

Participants spoke of having power in exercising choice, whether to discontinue texts or call a free phone for support services, not to feel pressured, and a conversational and friendly invitation to engage:

*"We're just checking in. You have the option if you want to kōrero [talk]. . . Not demanding to but implying it would be great to hear from you. Not saying, please do or you have to but we'd be happy if you did."*

The group emphasised "*mana*" [a definition used here is 'personal power'] of tangata whaiora, so they controlled decision-making in their recovery:

*"What if they don't want to talk to you? We're thinking about you and you're most welcome to call freephone 1737 if you want to korero about anything. I think the mana must stay. The choice must stay."*

As part of maintaining autonomy, participants wanted details about the text message schedule to be clear: who they were from, where they were sent from, how often, what would happen if they replied back to a text message and the process of opting out of receiving them.

## 2) Establishing a process of connection as a bridge to safety

From the first contact, creating a caring connection was viewed as important. Participants liked the idea of being called beforehand, to impart a sense of there being a real person behind the messages. This promoted feelings of being connected to someone who was holding them in mind, *"thinking about you,"* in a personal way:

*"It lets people know they haven't been forgotten, that contact's important. Yeah, and personalised. Being able to cater to someone who is from a certain age range who would respond a bit differently to different types of language use. . .If someone a lot younger talks a different way it might feel more personalised to their way of talking."*

The genuine and compassionate conveyance of hope resonated most with participants, feeling like someone cared:

*"I'm wondering who's this? Because I just had a really traumatic day. . .A bit on introduction to who is sending the text. . .Most government departments, they never write they care about you. I would love that, seeing that aroha nui [love] and thinking, oh they care. . .you've really hit that connection there."*

Participants recommended invitations to seek out support to be threaded in every text message. This consistency was a reminder that support was available to opt into and link to safety.

*"We're holding hope for you. This third text you can freephone text for support or kōrero - that might be nice with the first text you send. I don't know if they already know [freephone number]. . .this point needs to be in the first statement. . .That wasn't offered in the first one or the second one, the freephone. It just came over here in text seven."*

The frequency of the SMS schedule (delivered the day after their suicidal attempt, then one week, two weeks, three weeks, four weeks, six weeks and three months) was *"about right."* People would potentially be reminded of an event they wanted to move on from, evoking mixed feelings if it was untimely, if they were at work, or in a different headspace. No assumptions could be made about whether a text message would be interpreted positively or negatively:

*"If you hang out with your friends and your phone's on the bench and you get that, I mean, if you've got good enough mates they won't care. . . oh that's right, I tried to do that a few months ago. . .it's just the timing of it, maybe a little bit awkward. . . Might make a day not the best or make a bad day potentially worse but also the chance to make it significantly better."*

Some participants commented on the emotional tone of messages, *"that one felt warmer to me, got support and kōrero [talk] and she [the sender] wished me well. Thanks [name] for wishing me well, is probably what I would say after I read it, thanks."* They were receptive to this warmth, a sense of plugging into a human connection. They also highlighted that whaiora could opt out of receiving texts if they did not find them helpful.

### 3) Words as healing, rongoā

This theme emphasised i) the power of words and ii) Te Reo Māori (Māori language) as entry to te Ao Māori (the Māori world). Words were spoken of as having healing qualities like rongoā, a Māori term to describe healing through medicinal and other cultural treatments (one definition).

**i) The power of words.** One participant described how physical wounds could be seen but unseen wounds from a suicide attempt which required mending.

*"If you have the opportunity to hear somebody else talking and breathing life into you, it helps you grow and improve. . .this can be very beneficial because the healing will continue even after the attempt."*

Participants were sensitive to people feeling raw, traumatised and having negative memories of the event. They proposed using words that would accurately reflect and convey understanding, potentially that all may not be well:

*"If I've been in hospital because I've had a suicide attempt, someone a couple of days later saying, I hope your day's going well, I will feel like, do you really get it? You know, I'm probably still transitional. It sounds a little too happy to me."*

Carefully choosing words to frame messages was highlighted as very important particularly in the texts that people would receive in the few days after their suicide attempt.

*"It was good to meet you–just doesn't sound right. I mean, they've been there for a suicide attempt. . . something more like, we wanted to check in after our meeting yesterday."*

Subtle changes to the wording in texts were suggested to neutralise the tone of messages. Different phrases were trialled, for example:

*"I was just wondering how you are after yesterday, how are you going? I really like this one. . .because it's not insinuating anything. . .it's saying, hey I hope everything's going well, not necessarily saying anything's wrong. It's just a reminder, I think that's very middle ground. . . One word can change the perspective of the person completely."*

Participants were asked about visual images sitting alongside words in text messages, which they were cautiously receptive to:

*"You've got to be careful what images trigger for people. I mean, if they just tried to drown themselves, that [image of the water] wouldn't be it."*

They liked the idea of colours, seeing yellow as *"vibrant and hopeful,"* green as *"peaceful and chilled"* and purple as *"spiritual."*

**ii) Te Reo Māori (Māori language) providing entry to te Ao Māori (the Māori world).** Sending greetings in Te Reo Māori was seen as a gateway into te Ao Māori (the Māori world), which was also acknowledged as part of contemporary life in Aotearoa New Zealand:

*"Start with Tēnā koe [formal greeting] then go onto Kia ora [less formal greeting]."*

There were healing properties in introducing self as a means of whakawhanaungatanga, (a process of establishing relationships) to establish connection and rapport. Conveying *"I wish that you are well, I care about you"* in text messages epitomises connection. Participants advocated for translating key Māori words into English, to enhance access into Te Ao Māori (the Māori world) An example is the use of the word *mauri*, which one definition can mean vital essence or life force. This was translated further simply into energy.

*"It might not be the same but it would be good. Energy is a nice word. . .[next to] mauri. . .I feel that would be in everyday use and more widely accessible."*

Taking care in saying goodbye, even briefly, was considered important:

*"I think ngā mihi is great, that's commonly known. It's at the end of the message. . .you feel like okay, this is signing off. If I was feeling quite fragile and the message showed an understanding of how it was it might make me feel better."*

The use of Te Reo Māori was threaded throughout the messages, which further demonstrated the importance of language in connection for tangata whaiora Māori. Participants highlighted being mindful of a person's background and tailoring messages according to context and preferences, for example translating mauri into equivalent language for Pacific people.

## Discussion

The objective of this research was to design SMS text messages with peer support specialist staff working in mental health services, with attention to language, delivery and cultural context. We suggest three coherent principles to guide communication via texts: upholding a person's autonomy; establishing connection as a bridge to safety and words as healing rongoā. The third theme guides the design of messages that are tailored to indigenous culture in Aotearoa New Zealand, incorporating Te Reo Māori (Māori language) and providing entry to te Ao Māori (the Māori world).

The participants' experiential knowledge is central to this study. This research responds to a challenge of more closely examining the acceptability of interventions [12, 18], by working with people with a personal connection to suicide as vital partners in developing, testing and refining the text message series. This aligns with expressed expectations that health services will partner with service users and whānau [29]. The participants' embodiment of empowerment and hope represent the ultimate connection to choice and collaboration with mental health services [30]. Their contributions emphasise aspects of personal agency, such as intentionality, self-regulation and motivation [31]. As a collective, their focus on caring for the

person and capacity for reflectiveness undoubtably added value to the SMS series, promoting connection by use of caring language. The process of hearing the series of text messages spoken out loud was unifying and validating. The participants held in mind people receiving the texts, infusing compassion into the design process by the use of humanising language [32]. They heightened awareness of sensitive factors and contexts that surround people attempting suicide and returning to their lives thereafter. They endorsed self-efficacy [31], and the idea that even a simple message can have an important impact [5]. Establishing connection is the foundation of caring contacts, and the Māori concept of whakawhanaungatanga, which essentially focuses on relationships [33]. In this study, participants articulated the positive emotional impact of feeling genuinely cared for and not forgotten. This potentially enables an internal sense of safety and being in control, expressed in confidence to reach out for help, connecting to other support or opting out of the intervention.

The cultural context of this study is unique [34], in referring to words as healing rongoā, a solution that contributes to Māori health gains and reduces barriers to access and use of healthcare services [35]. This is exemplified by the discussion of *mauri* or life force that is evident in the vitality, integrity and energy within a person, and establishing positive relationships in the wider environment. Māori language is linked to the concept of *mana*, or dignity [36]. Participants from different cultural backgrounds connected with the expression of mauri as the essence of a person's character. The concept of mauri is important in understanding suicide [17], acknowledging the anguish and perplexity associated with suicide and the impacts on families, friends and whole communities. There was specific mention of healing the wounds of suicide. There is need to heal both our own wounds and the wounds of our lineage [15]. The task in preventing suicide is rejuvenating mauri, to overcome grief. The text messages express expectations for wellness. In essence, when we greet one another with 'Tēnā koe' and 'Kia ora', we are equally saying 'be well' [37]. Cultural identity and language, as means of establishing connection, are precursors to living well [38, 39]. Language was revised with humanistic and cultural qualities there were not necessarily clinical or technical. The participants highlighted whānau, or families, as prime agents in strengthening mauri [15]. This illustrates agency as integral to personal context and their wider sociocultural milieu [31].

## Strengths and limitations

A strength of this study is the unique sample, a large group of ethnically diverse people of different age ranges with lived experience who also work in publicly funded mental health services. The seventeen participants meet together on a regular basis. This familiarity enhanced rapport and open dialogue within the focus group. Members of the group had lived experience of mental health service involvement, and were familiar with navigating the mental health system but not all may have had lived experience of suicidal behaviour. Quotations are not individually attributed and the large size of the focus group may have privileged the voices of some group members, at the expense of those who were reticent, shy or least experienced. We do not consider these to be critical limitations in reporting the data. Their collective contributions are accepted as "taonga" (treasure) rather than "tokenistic [40]. We acknowledge the influence of our disciplinary origins and collective interpretations on the analytic narrative as a limitation. This was mitigated by independent co-coding by an experienced qualitative researcher and specific focus on cultural coding. The original text messages (Table 1) may reflect clinical psychiatry and psychology perspectives on the core components of caring contacts, being brief and non- demanding, whereas the final modified text highlights how the experiential knowledge of those with lived experience was elevated to convey a more humanistic tone, emphasising different modes of communication and nuances of Te Reo Māori. Peer specialist-

researchers were not involved in designing the original draft of the SMS text series and analysing the data, which would have provided a different perspective on the findings.

## Implications for practice and future research

This study illustrates the value of including people with lived experience of mental distress in planning and designing mental health interventions [18, 30]. Access to the voices of this important stakeholder group is facilitated by building trust in relationships with gatekeepers, who protect the welfare and interests of such groups. It would be helpful to involve peer support specialists in conducting research and implementing findings in healthcare services.

Text (SMS) messages can potentially overcome the limitations of other types of contact [6]. The optimal frequency and length of SMS contact may be up to 12 months, extending well beyond an acute crisis [1]. SMS can be considered an adjunct to formal follow up and service users can determine if it is conducive to their recovery by choosing to opt out of receiving messages. In terms of upscaling this type of caring contacts intervention, widespread use of mobile phone technology allows SMS to be readily deployed and may be more acceptable than alternative clinical therapeutic options, especially for young people [23, 41]. Text messages can be used as a brief intervention to identify emergency contacts during a suicide crisis [42] and promote engagement with treatment and safety planning. New Zealand census data indicates a high level of access to mobile phones with active internet connections [43]. Texts may be suitable for those who do not receive formal follow up from mental health services [44], although it cannot be assumed that all have access to, or funds to maintain mobile phones.

Emergency Departments are underutilised sites for suicide prevention for treating those who attempt suicide [8]. The ED can be a site to implement caring contacts as people discharged after a suicide attempt may not attend follow up. Obtaining perspectives of service users on engaging with caring contacts using technology would be a valuable focus for future qualitative research.

## Conclusion

Words are healing and powerful when delivered in a way that uphold a person's autonomy. People with lived experience breathe empowerment and hope into caring contacts interventions and should be considered vital partners in developing suicide prevention initiatives. The emotional impact of SMS text messages in feeling genuinely cared for promotes connection and may enable an internal sense of safety. There is benefit in designing this intervention with people who have lived experience. There is additional value in input from people from the same cultural background to tailor messages with more culturally nuanced language.

## Supporting information

**S1 File. Inclusivity in global research.**
(DOCX)

## Acknowledgments

The authors gratefully acknowledge the contributions of the research participants. Thank you to Moko Kairua for contributing her expertise, and to Rapua Whaioranga, Faletoa and Kiri Prentice for leading and supporting the study design. Thanks to Olivia High for her advice on the initial SMS series and to Lyn Lavery for her assistance with the analysis.

## Author Contributions

**Conceptualization:** Lillian Ng, Danielle Diamond, Mike Ang.

**Data curation:** Lillian Ng, Danielle Diamond, Mike Ang.

**Formal analysis:** Lillian Ng, Danielle Diamond, Mike Ang.

**Funding acquisition:** Lillian Ng, Danielle Diamond, Mike Ang.

**Investigation:** Lillian Ng, Danielle Diamond, Mike Ang.

**Methodology:** Lillian Ng, Danielle Diamond, Mike Ang.

**Project administration:** Lillian Ng, Danielle Diamond, Mike Ang.

**Resources:** Lillian Ng, Danielle Diamond, Mike Ang.

**Software:** Lillian Ng.

**Supervision:** Lillian Ng, Danielle Diamond, Mike Ang.

**Validation:** Lillian Ng, Danielle Diamond, Mike Ang.

**Visualization:** Lillian Ng, Danielle Diamond.

**Writing – original draft:** Lillian Ng, Danielle Diamond, Mike Ang.

**Writing – review & editing:** Lillian Ng, Danielle Diamond, Mike Ang.

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
