## [Decision Letter · Decision Letter 0]

4 Mar 2024

PONE-D-23-23584Text2whaiora after a suicide attempt: text message design alongside people with lived experiencePLOS ONE

Dear Dr. Ng,

Thank you for submitting your manuscript to PLOS ONE. After careful consideration, we feel that it has merit but does not fully meet PLOS ONE’s publication criteria as it currently stands. Therefore, we invite you to submit a revised version of the manuscript that addresses the points raised during the review process.

 Please submit your revised manuscript by Apr 18 2024 11:59PM. If you will need more time than this to complete your revisions, please reply to this message or contact the journal office at plosone@plos.org. Please include the following items when submitting your revised manuscript:A rebuttal letter that responds to each point raised by the academic editor and reviewer(s). You should upload this letter as a separate file labeled 'Response to Reviewers'.A marked-up copy of your manuscript that highlights changes made to the original version. You should upload this as a separate file labeled 'Revised Manuscript with Track Changes'.An unmarked version of your revised paper without tracked changes. You should upload this as a separate file labeled 'Manuscript'.

We look forward to receiving your revised manuscript.

Kind regards,

Karen M Davison, PhD

Academic Editor

PLOS ONE

Journal Requirements:

Did you know that depositing data in a repository is associated with up to a 25% citation advantage (https://doi.org/10.1371/journal.pone.0230416)? If you’ve not already done so, consider depositing your raw data in a repository to ensure your work is read, appreciated and cited by the largest possible audience. You’ll also earn an Accessible Data icon on your published paper if you deposit your data in any participating repository (https://plos.org/open-science/open-data/#accessible-data).

3. Please include a complete copy of PLOS’ questionnaire on inclusivity in global research in your revised manuscript. Our policy for research in this area aims to improve transparency in the reporting of research performed outside of researchers’ own country or community. The policy applies to researchers who have travelled to a different country to conduct research, research with Indigenous populations or their lands, and research on cultural artefacts. The questionnaire can also be requested at the journal’s discretion for any other submissions, even if these conditions are not met. 

Please find more information on the policy and a link to download a blank copy of the questionnaire here: https://journals.plos.org/plosone/s/best-practices-in-research-reporting. 

Please upload a completed version of your questionnaire as Supporting Information when you resubmit your manuscript.

5. Please note that funding information should not appear in the Acknowledgments section or other areas of your manuscript. We will only publish funding information present in the Funding Statement section of the online submission form. Please remove any funding-related text from the manuscript. 

6. We note that you have indicated that there are restrictions to data sharing for this study. For studies involving human research participant data or other sensitive data, we encourage authors to share de-identified or anonymized data. However, when data cannot be publicly shared for ethical reasons, we allow authors to make their data sets available upon request. For information on unacceptable data access restrictions, please see http://journals.plos.org/plosone/s/data-availability#loc-unacceptable-data-access-restrictions. 

7. Your ethics statement should only appear in the Methods section of your manuscript. If your ethics statement is written in any section besides the Methods, please move it to the Methods section and delete it from any other section. Please ensure that your ethics statement is included in your manuscript, as the ethics statement entered into the online submission form will not be published alongside your manuscript. 

**Additional Editor Comments:**

Thank you for your submissions and two reviews have been completed. I am hoping you can address their comments and resubmit your manuscript.

Kind Regards,

Karen Davison

Reviewers' comments:

Reviewer's Responses to Questions

**Comments to the Author**

1. Is the manuscript technically sound, and do the data support the conclusions?

Reviewer #1: Yes

Reviewer #2: Partly

2. Has the statistical analysis been performed appropriately and rigorously? 

Reviewer #1: N/A

Reviewer #2: N/A

3. Have the authors made all data underlying the findings in their manuscript fully available?

Reviewer #1: Yes

Reviewer #2: Yes

4. Is the manuscript presented in an intelligible fashion and written in standard English?

Reviewer #1: Yes

Reviewer #2: Yes

5. Review Comments to the Author

Reviewer #1: Overall this study is very important as a "method" SMS text messaging for structurally vulnerable people who experience mental health challenges and or mental illness. The value added of this qualitative study is in the inclusion of peers as evaluators for text messages.

1.The background shows a good understanding of the literature and need for the study. However, it would be good to situate the study within a global context. For example, the human rights of Indigenous people globally and the global movement toward "Patient Oriented Research" or the inclusion of patient involvement in research extends to public involvement, which means that research is carried out ‘with’ or ‘by’ members of the public rather than ‘to’, ‘about’ or ‘for’ them. There is another important concept of "peer" and this has not been defined interms of where this concept came from? this again might help contextualize the paper and its findings within a broader audience.

2. On Page p.4 can you clarify which sentence is requiring a page number reference? see: p2.” Meta-analytic studies show caring contacts reduce some suicidal behaviours and there is limited evidence in reducing suicide mortality, hospitalisation or presentations to the emergency department (Skopp et al., 2023).

3.On page p. 5 please elaborate on how this affected access and support see "…For example, a New Zealand qualitative study asked people with suicidal ideation or suicidal behaviour about their emotional responses to caring contacts and accessing supports (High, 2022). SMS text messages were preferred, compared with other forms of follow up".

4.Methods: If you are using reflexivity as a method of analysis vs a “ method” please elaborate. You could describe this method as a qualitative method incorporating critical reflexivity?- semi structured interviews peer involved research..is not as clear. You note that you adapt from Braun and Clarke to use thematic reflexive analysis, how critical is it that your peers had lived experience, when some research participants did not have lived experience of suicide? this leads to the rich texts and quotes which came from your participants, and it is difficult to see who said what.

The authors mention on p.7 "The analytic narrative of the research was influenced by the intersections of our disciplinary origins and collective interpretations". However as a reader it is hard to see the variation of your participant responses in relation to the themes. The collective meaning that was derived makes sense but we don't get a clear sense of who said what and why.

5. The recommendation under number 4 is to provide some distinctions amongst the data participants as later the authors go on to state on p. 17 " Culturally nuanced findings may be transferable internationally, where there are indigenous or first nations peoples who are familiar with the Turamarama declaration, which affirms the agency and rights of these communities to find their own solutions for health, including suicide prevention". It is recommended that these kinds of human rights and Indigenous rights policies be addressed in the backround section as well as this should not be written as a limitation- given you adopt Braun and Clarke's thematic and reflexive analysis which foreground the active role of the researcher in developing and reflecting on the findings. This idea might also be strengthened if the authors clarify the role of psychiatrist and clinical psychologist. If embedded in a culturally safe framework what is the role of these professionals?

6. More analysis could be provided related to frequency of text schedules as most participants noted that they would prefer to have messages stopped after 3 months, but connect and support should last longer? so there maybe a need to explain what this means in relation to acute mental health crisis and need for follow up support.

7. Not clear how NVivo data analysis software program was used in the analysis or storing the data, please use standard spelling. Limitations could include a mention of other methods that could be used in future to strengthen "peers" voices.

8. The discussion and conclusion provide good implications for policy recommendations. This section could be strengthened by highlighting key policy and practice recommendations for institutional settings, i.e. main stream or Maori mental health settings where mental health supports are provided for patients and families that have experienced a suicide attempt and/or lived experience of someone they know. Is there also a need to adopt an intersectional analysis for diversity of needs that attend to having hand held "devices" where affordibility might be an issue. This section could be strengthened with a discussion related to policy and practice recommendations keeping in mind the peer recommendations.

9. Finally, there is good attention to power but less attention to individual agency and the factors that promote agency. The authors might consider this to add to the nuanced discussion on power. Further, what do the findings mean globally related to issues about equity and virtual care technologies, again this could also be added in the background sections.

The paper could be strengthened with attention to these details.

Reviewer #2: This is an interesting and important study. It is well written and clearly presented.

At the same time, the manuscript can be enhanced through attention to the following:

1. How is 'lived experience' being conceptualized in this study?It would be good for the authors to elaborate on this and cite the literature that acknowledges the limitations of singular and static categorical identities, which do not fully recognize that people often occupy multiple identity categories, including clinician/service user/person with lived experience/suicide survivor/professional, etc. See for example J. Voronka's work on the politics of people with lived experience

2. It would be illuminating to see the original text messages prepared by the psychiatrist/psychologist, which were provided to the focus group for evaluation. Note that this relegates the persons with lived experience to the role of 'editors' more than co-designers of the text messages. I wonder if this might be considered as a potential limitation.

3. In the description of the analysis, the authors note that they were influenced by their 'disciplinary origins' but this is not made explicit. What is the theoretical orientation that is being brought to bear on these data?

4. How is the experiential knowledge of those with lived experience being elevated/mobilized to add value to what is already known about the content/form of caring contacts?

5. The qualitative data provided in support of the claims is often rather sparse and the analysis would be strengthened if there were more in-depth quotations provided across the participants

6. What was the length of time provided for the focus group? For 17 people to be able to share their views, there would need to be ample time given to listening to the diverse range of perspectives.

6. PLOS authors have the option to publish the peer review history of their article (what does this mean?). If published, this will include your full peer review and any attached files.

Reviewer #1: No

Reviewer #2: No

---

## [Author Response · Author response to Decision Letter 0]

5 Apr 2024

Journal Requirements:

The revised manuscript and title page are now formatted to meet the PLOS ONE style requirements.

Did you know that depositing data in a repository is associated with up to a 25% citation advantage (https://doi.org/10.1371/journal.pone.0230416)? If you’ve not already done so, consider depositing your raw data in a repository to ensure your work is read, appreciated and cited by the largest possible audience. You’ll also earn an Accessible Data icon on your published paper if you deposit your data in any participating repository (https://plos.org/open-science/open-data/#accessible-data).

The authors have elected not to deposit the focus group data, as it contains some participants’ more sensitive and personal information.

3. Please include a complete copy of PLOS’ questionnaire on inclusivity in global research in your revised manuscript. Our policy for research in this area aims to improve transparency in the reporting of research performed outside of researchers’ own country or community. The policy applies to researchers who have travelled to a different country to conduct research, research with Indigenous populations or their lands, and research on cultural artefacts. The questionnaire can also be requested at the journal’s discretion for any other submissions, even if these conditions are not met. 

Please find more information on the policy and a link to download a blank copy of the questionnaire here: https://journals.plos.org/plosone/s/best-practices-in-research-reporting. 

Please upload a completed version of your questionnaire as Supporting Information when you resubmit your manuscript.

The PLOS questionnaire on inclusivity in global research has been completed and is attached as supporting information.

The funding information and financial disclosure sections are completed.

5. Please note that funding information should not appear in the Acknowledgments section or other areas of your manuscript. We will only publish funding information present in the Funding Statement section of the online submission form. Please remove any funding-related text from the manuscript. 

There is no reference to funding in the manuscript. 

6. We note that you have indicated that there are restrictions to data sharing for this study. For studies involving human research participant data or other sensitive data, we encourage authors to share de-identified or anonymized data. However, when data cannot be publicly shared for ethical reasons, we allow authors to make their data sets available upon request. For information on unacceptable data access restrictions, please see http://journals.plos.org/plosone/s/data-availability#loc-unacceptable-data-access-restrictions. 

Excerpts of the transcripts relevant to the study will be made available upon reasonable request from the corresponding author.

7. Your ethics statement should only appear in the Methods section of your manuscript. If your ethics statement is written in any section besides the Methods, please move it to the Methods section and delete it from any other section. Please ensure that your ethics statement is included in your manuscript, as the ethics statement entered into the online submission form will not be published alongside your manuscript. 

The ethics statement is located in the Methods section, as per the online submission form.

Additional Editor Comments:

Thank you for your submissions and two reviews have been completed. I am hoping you can address their comments and resubmit your manuscript.

Kind Regards,

Karen Davison

Reviewers' comments:

Reviewer's Responses to Questions

Comments to the Author

1. Is the manuscript technically sound, and do the data support the conclusions?

Reviewer #1: Yes

Reviewer #2: Partly

2. Has the statistical analysis been performed appropriately and rigorously? 

Reviewer #1: N/A

Reviewer #2: N/A

3. Have the authors made all data underlying the findings in their manuscript fully available?

Reviewer #1: Yes

Reviewer #2: Yes

4. Is the manuscript presented in an intelligible fashion and written in standard English?

Reviewer #1: Yes

Reviewer #2: Yes

5. Review Comments to the Author

Reviewer #1: Overall this study is very important as a "method" SMS text messaging for structurally vulnerable people who experience mental health challenges and or mental illness. The value added of this qualitative study is in the inclusion of peers as evaluators for text messages.

1.The background shows a good understanding of the literature and need for the study. However, it would be good to situate the study within a global context. For example, the human rights of Indigenous people globally and the global movement toward "Patient Oriented Research" or the inclusion of patient involvement in research extends to public involvement, which means that research is carried out ‘with’ or ‘by’ members of the public rather than ‘to’, ‘about’ or ‘for’ them. There is another important concept of "peer" and this has not been defined interms of where this concept came from? this again might help contextualize the paper and its findings within a broader audience.

We have added to the introduction:

“This is in keeping with the movement toward patient-oriented research, that aims to empower patients in the process of research, optimise research design, enhance validity and make knowledge exchange more effective. (19). The goal of patient-oriented research is achieved when research findings improve the health of the population under study (20,21).Having lived experience of a condition, of navigating the healthcare system and being willing to share those experiences are important contributions.”

“We describe our peer specialist participants as “people with lived experiences,” and acknowledge that this term encapsulates a range of experiences such as distress and a range of social stressors (24). We take “lived experience” to mean someone who has an experience of mental distress, and acknowledge their diverse experiences, and intersections with other identities.”

Further details are provided in the recruitment section: “All had lived experience of mental health care and most had direct experience of suicidal behaviour, such as suicidal ideation, a suicide attempt or the death of a family member from suicide. However, all had experience of supporting someone with suicidal behaviour.”

2. On Page p.4 can you clarify which sentence is requiring a page number reference? see: p2.” Meta-analytic studies show caring contacts reduce some suicidal behaviours and there is limited evidence in reducing suicide mortality, hospitalisation or presentations to the emergency department (Skopp et al., 2023).

The page number referred to the Josifovski, 2022 article in the sentence before. We have seen fit to condense the quote from the article and remove the page number, as it appears as a distraction.

3.On page p. 5 please elaborate on how this affected access and support see "…For example, a New Zealand qualitative study asked people with suicidal ideation or suicidal behaviour about their emotional responses to caring contacts and accessing supports (High, 2022). SMS text messages were preferred, compared with other forms of follow up".

This sentence is not clear as it makes two points. We now make one point and have modified the next two sentences, which briefly describe the two studies.

“These typically involve small numbers (22,23). Larsen et al’s research co-designed a SMS text message brief contact intervention with lived experienced groups for delivery in an ED setting after a suicide attempt. In another study, people with suicidal ideation or suicidal behaviour preferred SMS text messages, compared with other forms of follow up (22).”

4.Methods: If you are using reflexivity as a method of analysis vs a “ method” please elaborate. You could describe this method as a qualitative method incorporating critical reflexivity?- semi structured interviews peer involved research..is not as clear. 

Reflexivity is part of the methods, and we acknowledge our backgrounds as influencing the research process. We have changed the heading to “Research context” which more accurately describes the earlier part of the section. While reflexivity is not explicit as a heading, we have integrated this lens into the manuscript; for example we note the reviewer’s comment re “analytic narrative" and have placed this in the limitations section. 

You note that you adapt from Braun and Clarke to use thematic reflexive analysis, how critical is it that your peers had lived experience, when some research participants did not have lived experience of suicide? this leads to the rich texts and quotes which came from your participants, and it is difficult to see who said what. The authors mention on p.7 "The analytic narrative of the research was influenced by the intersections of our disciplinary origins and collective interpretations". However as a reader it is hard to see the variation of your participant responses in relation to the themes. The collective meaning that was derived makes sense but we don't get a clear sense of who said what and why.

We have modified this sentence slightly:

”All had lived experience of mental health care and most had direct experience of suicidal behaviour, such as suicidal ideation, a suicide attempt or the death of a family member from suicide. However, all had experience of supporting someone with suicidal behaviour.”

 We have added that “quotations are not individually attributed” and the “collective contributions” of participants which “are accepted as “taonga” (treasure) as opposed to being “tokenistic.” (Ross et al, 2023).” 

We have shifted the sentence: “The analytic narrative of the research was influenced by the intersections of our disciplinary origins and collective interpretations.” To the Strengths and Limitations section and modified it to: 

“We acknowledge the influence of our disciplinary origins and collective interpretations on the analytic narrative as a limitation. This was mitigated by independent co-coding by an experienced qualitative researcher and specific focus on cultural coding. The original text messages (Table 1) may reflect a clinical psychiatry and psychology perspective on the core components of caring contacts, being brief and non- demanding, whereas the final modified text highlights how the experiential knowledge of those with lived experience was elevated to convey a more humanistic tone, emphasising different modes of communication and nuances of Te Reo. Peer specialist-researchers were not involved in designing the original draft of the SMS text series and analysing the data, which would have provided a different perspective on the findings.”

5. The recommendation under number 4 is to provide some distinctions amongst the data participants as later the authors go on to state on p. 17 " Culturally nuanced findings may be transferable internationally, where there are indigenous or first nations peoples who are familiar with the Turamarama declaration, which affirms the agency and rights of these communities to find their own solutions for health, including suicide prevention". It is recommended that these kinds of human rights and Indigenous rights policies be addressed in the backround section as well as this should not be written as a limitation- given you adopt Braun and Clarke's thematic and reflexive analysis which foreground the active role of the researcher in developing and reflecting on the findings. This idea might also be strengthened if the authors clarify the role of psychiatrist and clinical psychologist. If embedded in a culturally safe framework what is the role of these professionals?

The sentence " Culturally nuanced findings …suicide prevention” has been moved to section on cultural relevance to indigneous peoples in the introduction.

We have retained the general background of the research team but have added some more information: 

One of the authors (DD), is a psychologist of Māori (Kāi Tahu and Ngāpuhi) descent, and was instrumental in leading the research team in Māori and Pacific consultation to ensure the study incorporated 

---

## [Decision Letter · Decision Letter 1]

25 Jun 2024

Text2whaiora after a suicide attempt: text message design alongside people with lived experience

PONE-D-23-23584R1

Dear Dr. Ng,

We’re pleased to inform you that your manuscript has been judged scientifically suitable for publication and will be formally accepted for publication once it meets all outstanding technical requirements.

Kind regards,

Karen M Davison, PhD

Academic Editor

PLOS ONE

Additional Editor Comments (optional):

Based on review of your responses to the reviewers and the revised manuscript I recommend acceptance. My only suggestion is to add to the quotations some reference to the participant to show if there was a diversity of quotes e.g., participant #1, participant #2.

Reviewers' comments:

Reviewer's Responses to Questions

**Comments to the Author**

1. If the authors have adequately addressed your comments raised in a previous round of review and you feel that this manuscript is now acceptable for publication, you may indicate that here to bypass the “Comments to the Author” section, enter your conflict of interest statement in the “Confidential to Editor” section, and submit your "Accept" recommendation.

Reviewer #2: All comments have been addressed

2. Is the manuscript technically sound, and do the data support the conclusions?

Reviewer #2: (No Response)

3. Has the statistical analysis been performed appropriately and rigorously? 

Reviewer #2: (No Response)

4. Have the authors made all data underlying the findings in their manuscript fully available?

Reviewer #2: (No Response)

5. Is the manuscript presented in an intelligible fashion and written in standard English?

Reviewer #2: (No Response)

6. Review Comments to the Author

Reviewer #2: (No Response)

7. PLOS authors have the option to publish the peer review history of their article (what does this mean?). If published, this will include your full peer review and any attached files.

Reviewer #2: No

---

## [Editor Report · Acceptance letter]

1 Jul 2024

PONE-D-23-23584R1 

PLOS ONE

Dear Dr. Ng, 

I'm pleased to inform you that your manuscript has been deemed suitable for publication in PLOS ONE. Congratulations! Your manuscript is now being handed over to our production team.

Kind regards, 

on behalf of

Dr. Karen M Davison 

Academic Editor

PLOS ONE